# Evaluating and Modeling the Degradation of PLA/PHB Fabrics in Marine Water

**DOI:** 10.3390/polym15010082

**Published:** 2022-12-25

**Authors:** Qi Bao, Ziheng Zhang, Heng Luo, Xiaoming Tao

**Affiliations:** 1Research Institute of Intelligent Wearable Systems, The Hong Kong Polytechnic University, Hong Kong 999077, China; 2School of Fashion and Textiles, The Hong Kong Polytechnic University, Hong Kong 999077, China

**Keywords:** PLA, PHB, marine plastics pollution, artificial intelligence, neural network

## Abstract

Developing degradable bio-plastics has been considered feasible to lessen marine plastic pollution. However, unanimity is still elusive regarding the actual degradability of bio-plastics such as polylactide (PLA) and poly(hydroxybutyrate) (PHB). Thus, herein, we studied the degradability of fabrics made from PLA/PHB blends in marine seawater. The dry-mass percentage of the PLA/PHB fabrics decreased progressively from 100% to 85~90% after eight weeks of immersion. Two environmental aging parameters (UV irradiation and aerating) were also confirmed to accelerate the abiotic hydrolysis of the incubated fabrics. The variation in the molecular structure of the PLA/PHB polymers after the degradation process was investigated by electrospray ionization mass spectrometry (ESI-MS). However, the hydrolysis degradability of bulky PLA/PHB blends, which were used to produce such PLA/PHB fabrics, was negligible under identical conditions. There was no mass loss in these solid PLA/PHB plastics except for a decrease in their tensile strength. Finally, a deep learning artificial neural network model was proposed to model and predict the nonlinear abiotic hydrolysis behavior of PLA/PHB fabrics. The degradability of PLA/PHB fabrics in marine water under the synergistic destructive effects of seawater, UV, and dissolved oxygen provides a pathway for more sustainable textile fibers and apparel products.

## 1. Introduction

Marine pollution derived from fossil-fuel-based plastics is emerging as a public hazard to global ecosystems and human health [1]. This is mainly attributed to (a) these plastics’ persistent lifetime (e.g., 600 years for fishing line, 450 years for beverage bottles) [2]; (b) their large-scale implementation in a wide range of applications including packaging (35.9%), construction (16.0%), textiles (14.5%), and consumer goods (10.3%); and (c) their cumulative accumulation over decades [3]. Furthermore, their manufacturing process also produces a large amount of carbon-dioxide-equivalent (10~13%) global greenhouse gas emissions, which exacerbates another crisis, i.e., climate change [4]. As an ever-growing concern, microplastics (typically defined as < 5 mm in at least one dimension), which have attracted much more attention recently, have already spread ubiquitously from the Pacific Ocean’s Mariana Trench to the Antarctic iceberg [5]. These microplastics impose a potential threat to marine fish, animals, and human beings via food chain transfer, causing harmful toxicological and/or physiological impairments ranging from DNA damage, altered metabolism, inflammation, decreased growth, reduced cognitive function, reproductive harm, and mortality [6]. The principal form of microplastics is microfiber (84~85%) [7,8]. Synthetic textiles are one of the major sources of such microplastic pollution, contributing approximately 35% of the global release of primary microplastics [9].

To mitigate marine (micro)plastic pollution, there is an urgent need to develop a series of efficient solutions and methods. Undeniably, these common pervasive plastics are indispensable in mankind’s daily life, although their disadvantages have already been studied systematically by pioneer researchers. To address this problem, one feasible and economical resolution has been proposed, which is exploring bio-plastics/bio-polymers as a greener alternative to the non-degradable fossil plastics. Generally, these bio-plastics/bio-polymers are manufactured from renewable or recycled raw bio-based materials or from biological processes with a low carbon footprint, enabling a more sustainable economy [10]. These kinds of bio-plastics include poly(butylene succinate) (PBS), poly(ε-caprolactone) (PCL), poly(butylene adipate-co-teraphthalate (PBAT), etc. Among them, PLA holds the largest market share worldwide in 2022, accounting for 20.7% of the total bio-plastics calculated on the production capacities [11]. Another well-known bio-plastic, poly(hydroxybutyrate) (PHB), as the major member of polyhydroxyalkanoates (PHA), has also attracted tremendous attention, accounting for 3.9% of the global production capacities [12,13]. However, both PLA and PHB are semi-crystalline and brittle. After blending PLA with optimized mass ratios of PHB ranging from 20% to 40%, the mechanical properties of PLA, in particular the mechanical resistance toward extension and impact, have been improved significantly [13,14,15]. Thus, the PLA/PHB bio-polymers have shown great promise in applications to textiles and clothing in sight of their good biodegradability, hydrophilicity, biocompatibility, and recyclability.

However, the viewpoints of researchers are divided on whether the degradability of these bio-polymers such as PLA and PHB is effective in practical fields, such as in marine seawater. For example, Tsuji et al. reported that no degradation was found regarding PLA films after immersion in the seawater of the Pacific Ocean at 25 °C for ten weeks [16]. The degradation rate of PHB was merely 9%. Not to be outdone, Bagheri et al. also reported that PCL and PLA did not degrade at all, and approximately 8% degradation was observed for PHB after one year [17]. We recently reported such abiotic hydrolysis degradation behavior of PLA/PHB in the form of a mature-end textile product scale in confined marine seawater [18]. To promote their large-scale implementation, more controversies still require further elucidation. For example, another key issue behind the public outcry is that there is not yet an appropriate mathematical model to fit the degradation process in a practical environment, assuming these bio-plastics are eventually degradable in marine seawater. This also results from the difficult fact that the overall experimental data are still scant. The scale of relevant scientific data is extremely limited. Beyond question, various environmental parameters, except for microbes, also play a vital role in the degradation process of these bio-plastics products, and their effects also require further clarification. Meanwhile, the emergence of artificial intelligence (AI) technology, in particular the application of AI algorithms, brings new opportunities to process complex data such as protein folding prediction [19], computer vision [20], natural language processing, and so on [21]. This technology also provides a contemporary perspective and an innovative methodology for predicting complex tasks. Herein, an AI algorithm model has been adapted to analyze the marine degradation process of the PLA/PHB fabrics under parallel conditions to obtain the experimental law on the effects of environmental parameters.

In this work, both bulky PLA/PHB plastics in the form of raw materials and their mature textile fabric products have been studied together to investigate the similarity and differences in the degradation behaviors in parallel conditions to exclude the interference of temporal and spatial parameters. The variation in mass loss and mechanical properties, as well as “soft” electrospray ionization mass spectrometry (ESI-MS) spectra, has been applied to make the comparison between degradation results as well as the molecular mechanisms involved. Technically, the mathematical description of these so-called “degradable bio-polymers” such as PLA and PHA is of vital significance in evaluating the degradation process of end-products such as fabrics and clothing, particularly during the periods after their disposal into marine ecosystems. Slower degradation means a higher risk of being devoured by marine life such as fish, turtles, or seabirds. However, faster degradation means less sustainability as far as a circular economy is concerned. This requires an ingenious balance between these two aspects [22]. The prime priority lies in the prerequisite that we could acquire accurate degradation prediction by using experimental data. Thereafter, an advanced deep-learning methodology, i.e., an artificial neural network (ANN), is introduced to calculate the environmental effects of selective multi-parameters and to fit/predict the degradation process of the PLA/PHB textiles in environmental conditions for predetermined periods as a versatile and expandable scientific model.

## 2. Materials and Methods

### 2.1. Materials and Characterization

Both PLA/PHB granules and multi-filament yarns were obtained from Ningbo Hesu Fibers Co., Ltd. (Yuzhao City, China) [23]. The blend ratio of PLA/PHB was fixed at 70/30 for the melt-spinning process to produce fully drawn filament yarns. The linear density of the filament was 75D/48F, where D is Denier (mass density as gram per 9000 m) and F is the number of filaments in an FDY yarn. The knitted fabrics were made by using a circular knitting machine (WUXI ERVA Knitted Fashion Co. Ltd., Wuxi City, China). A gauge of 28 needles per inch was adopted for producing single jersey-knitted fabrics from the PLA/PHB filament yarns. The filament yarns were fed into the knitting machine directly, and the pretension used was 2~3 cN. The initial diameter of each PLA/PHB fiber was averaged to 20 μm. The bulky PLA/PHB plastics were made from PLA/PHB granules. The PLA/PHB granules were melted in a PTFE mold at 190 °C under vacuum for 3 h and cooled naturally. These pie-shaped PLA/PHB bulky samples featured a diameter of 55.70 mm and a thickness of 4.79 mm.

The tenacity, elongation, Young’s modulus, and extension energy of the PLA/PHB bulky samples were obtained from an Instron 5566 universal testing machine (Norwood, MA, USA). All the tensile tests were carried out in reference to the ASTM D638 standard. Before measurement, all PLA/PHB bulky samples were conditioned in a vacuum oven for 48 h at 60 °C. Tenacity at break and percent elongation at break were calculated automatically from tension data.

The degradation tests of the PLA/PHB yarns, fabrics, and bulky samples were carried out in separate natural seawater baths (500 mL). The degradation tests were performed under four unique conditions: immersion in (ⅰ) static natural seawater, (ⅱ) aerobic natural seawater (all the air flow rate was set to be 4.3 ± 1.0 SLPM) under a dark box, (ⅲ) static natural seawater under ultra-violet light (low-pressure mercury-vapor fluorescent lamps, TL-D 18 W × 3, 370 nm, Philips Lighting, Amsterdam, Netherlands) in a specific home-made UV chamber, and (ⅳ) aerobic natural seawater under ultra-violet light in the same UV chamber. All the seawater was renewed every week.

The structural integrity of the aged PLA/PHB products was investigated by electrospray ionization mass spectrometry (ESI-MS). Small pieces of aged PLA/PHB fabrics (~2.5 mg) were dissolved and collected in 0.7 mL chloroform (HPLC grade) solvent. Then the samples were analyzed with high-performance liquid chromatography (HPLC, 1290 Infinity, Agilent, Santa Clara, CA, USA) coupled with liquid chromatography (LC) electron spray ionization (ESI) source and an accurate-mass quadrupole time-of-flight mass spectrometer (Q-TOF/MS, 6540, Agilent, Santa Clara, CA, USA). The mass spectrometer was operated in negative mode, with the detected mass in the range from 50 to 10,000 atomic mass units. The HPLC-grade mixed solvent of acetonitrile/chloroform (1:1, *v*:*v*) was adopted as the mobile phase. Experimental data were processed and deconvoluted using Agilent MassHunter 10.0 qualitative software(Santa Clara, CA, USA).

### 2.2. Artificial Neural Network (ANN) Model

The data-fitting and prediction of degradation behavior in terms of the mass-loss percentage of PLA/PHB fabrics were carried out by the implementation of an artificial neural network (ANN) model in this work. Thanks to Hornik’s theoretical work, ANNs can act as approximate Borel measurable functions, which means these ANNs can be used to replace sophisticated data processing functions instead of intensive labor in deriving close expressions of formulae, which sometimes never explicitly exist at all [24,25,26,27]. All ANN calculations were carried out on TensorFlow 2.9.1 (Google, Mountain View, CA, USA) and Scikit-Learn 1.1 (USA) mathematical software architecture. In this study, a three-layer deep-learning neural network featuring rectified linear unit (ReLU) activation functions for the first two layers and a sigmoidal activation function for the last layer was built and designed for the ANN model training, model testing, and model implementation. The back-propagation algorithm was used to update/finetune the weights and biases variants present in this network. The mean square error (MSE) was selected as the loss function, which measures the performance of the ANN network according to the following equation:(1)$$MSE=\frac{1}{n}\sum_{j=1}^{n}{\left(y_{j}-{ŷ}_{j}\right)}^{2},$$
where *n* is the number of total data points; *y_j_* represents the actual value of the output layer, i.e., the mass-loss percentage of PLA/PHB fabrics after *j* days’ immersion; *ŷ_j_* is expressed as the predicted value of the mass-loss percentage of PLA/PHB fabrics after *n* days’ immersion; and *j* is an index of data.

Meanwhile, the mean absolute error (MAE) was selected as another monitoring metric, as defined by the following equation:(2)$$MAE=\frac{1}{n}\sum_{j=1}^{n}\left|y_{j}-{ŷ}_{j}\right|,$$

The ANN model training and testing platform consist of a Windows 11 Pro 64-bit operating system, and the hardware system includes a 2.5 GHz Core i9 11900 CPU (Intel, Mountain View, CA, USA), 32 GB DDR4 memory, and an RTX 3060 GPU graphics card (Nvidia, Santa Clara, CA, USA).

## 3. Results and Discussion

The experimental seawater was collected from the surface seawater of Kowloon Bay (Hum Hung, Hong Kong, China), located at latitude 22°30′ N and longitude 114°18′ E, as shown in the geographical map in Figure 1A. Hong Kong is a coastal city adjacent to the South China Sea, located in the subtropical climate zone. The annual average temperature of seawater is 24.2 °C, varying between 19.0 °C and 29.0 °C in the year 2021. The average salinity, dissolved oxygen (mg/L), pH value, 5-day biochemical oxygen demand (mg/L), and number of *E*. *coli* (CFU/100 mL) was 30.9 (27.4~33.1), 5.6 (4.5~6.6), 7.9 (7.6~8.1), 0.8 (0.3~2.0) and 490 (79~5500), respectively. The degradation experiments were carried out during the period from September to November. The detailed multiple physicochemical properties of the seawater were monitored monthly in a comprehensive marine water quality monitoring system, as depicted in Figure 1B–G. The sampled seawater was applied directly without any further purification. The variety of the pH value and the temperature, as well as the microbes, was not so significant during the experimental period in terms of such subtropical seawater. In this work, the logic numbers 1 and 0 have been used to represent the cases with and without UV for simplicity, given that the complexity of UV levels was out of the range of this work. The average saturated dissolved oxygen (DO) value of the surface seawater in the sampling site of Hong Kong in 2021 was in the range of 7.0~7.1 mg/L. The average saturation percentage of DO of the sampling seawater in 2021 was 79%. Similarly, the logic numbers 1 and 0 have been used to represent the cases with and without aeration for simplicity. In practice, various environmental parameters might contribute to accelerating/slowing the degradation of PLA/PHB fabrics, which include but are not limited to UV irradiation and dissolved oxygen.

As a facile but accurate quantitative method, the gravimetric mass-loss percentage has been adopted to determine the degradation rate of PLA/PHB fabrics in this work and was defined by the following equation. Three parallel samples with different physical sizes (3 cm × 5 cm, 4 cm × 4 cm, and 5 cm × 5 cm) were weighed with a precise five-digit balance and the average values are reported here. The according PLA/PHB fabrics samples were labeled as PLAHBA15, PLAHBA16, and PLAHBA25, respectively.
(3)$$Degradation\;percentage\;\left(\%\right)=\frac{m_{n}-m_{0}}{m_{0}}\times 100,\;$$
where *m_n_* is the dry-weight value of the PLA/PHB fabrics after *n* weeks of immersion in seawater, and *m*_0_ is the initial dry-weight value of the fresh PLA/PHB fabrics before immersion in seawater.

As one of the valuable forecast tools for the handling and prediction of big data on a large scale, artificial neural network (ANN) modeling has been successfully implemented in many fields of environmental science and engineering. Meanwhile, an ANN is also a powerful method for multivariate data analysis due to its reliable, robust, and salient capability and flexibility for capturing the non-linear relationships between variables (multi-inputs/outputs) in multivariate systems. Herein, ANN modeling has been introduced to fit and predict the mass-loss degradation process of PLA/PHB fabrics, in which case the continuity of the mass loss was much in self-evidence. As shown in Figure 2, a neural network was built, consisting of two hidden layers with ReLU activation functions called neurons and the last layer of one output with sigmoid transfer functions. The mathematical strength of these interconnections was determined by the weight associated with the neurons as well as additional bias. The ANN model was trained according to the backpropagation algorithm, which minimized the MSE loss between the real output (*y*) and the predicted output (*ŷ*) in the output layer. The Stochastic Gradient Descent (SGD) was applied as the optimization algorithm. Then, the nonlinear logical regression relationships between the multiple input variables and the output variable were set up based on such an ANN model after hundreds of epochs of the data-training process. Herein, the multiple input variants of the neural network were selected after data feature engineering and the multiple input variants included the physical size of the fabrics (cm^2^), the immersion duration time (weeks), the UV radiation, and the air bubbling (DO), in this case. Among them, the physical size of the fabrics (cm^2^) and the immersion duration time (weeks) were the normalized numeric variables, while the UV radiation and the air bubbling (DO) were categorical variables. The hyperparameter architecture of the neural networks, i.e., the number of layers of the ANN model, the number of neurons per layer, and the activation functions, were heuristically determined based on the cross-validation algorithm, where the whole dataset was split into the testing sub-set (one fifth) and the training sub-set (four fifths), as shown in Figure 3.

As shown in Figure 4, the distinct mass-loss phenomena have been identified in all cases of PLA/PHB samples after immersion, yet the degradation rate varied under different environmental parameters over time. The degradation percentage after 8 weeks’ immersion was in the range of 10.25 wt.%~16.26 wt.%. Generally, the actual mass-loss rate of the PLA/PHB fabrics was fast at the initial stage (0~2 weeks) of the whole experimental period. Afterward, the experimental degradation rate of the PLA/PHB fabrics slowed down gradually in the following stage (2~8 weeks). The total mass loss of the PLA/PHB fabrics might be attributed to the diffusion of water-soluble small molecules within the PLA/PHB fibers and/or the hydrolysis degradation effect of seawater. Such comparative experimental results revealed that the degradation rates of PLA/PHB fabrics under four different environmental conditions follow the order: case (UV + DO) > case (UV) >> case (DO) > case (static seawater). By comparison, the difference between the sample of (A) static seawater and the sample of (B) aerated seawater was not obvious, as was the difference between the sample of (C) UV-lighting static seawater and the sample of (D) UV-lighting aerated seawater. The presence of UV light accelerated the hydrolysis degradation process of PLA/PHB fabrics, as can be seen by comparing the data of Figure 4A,C. The mass-loss percentage of the two PLA/PHB fabrics in the presence of UV light was higher than that of the two samples in the absence of UV light. However, the accelerating effect of dissolved air was smaller than that of UV exposure in terms of mass-percentage values. All the detailed raw data and predicted data are summarized in the Appendix A.

The MSE value after 1000 epochs was decreased to a minimum of 0.000343, which is close to zero, when seven neurons were optimized and implemented within the ANN model. The MSE value gradually became steady after 159 epochs, as shown in Figure 5. At the same time, the other monitoring index, the MAE value, was also reduced to a minimum of 0.0143 after 1000 epochs. The decrease in the MAE value gradually became steady after 164 epochs. More importantly, no obvious overfitting phenomena were found during the data-training process. Meanwhile, the contribution of surface size was negligible among the four input variables explored and discovered by the ANN model after the data training process. There was no difference in the degradation of PLA/PHB fabrics with different physical sizes.

Figure 6 shows the apparent geometry of the sustainable PLA/PHB fabrics before and after different conditions. The fresh PLA/PHB fabrics exhibited a tight structure before the aging experiments. After eight weeks of aging, the integrity of the three cases (static seawater, UV, or DO) remained basically intact, although the structure seemed loose. However, in the presence of both UV and DO aging, the status of such a case was the most serious among these cases. The fabric integrity deteriorated and a visible hole through the center of the PLA/PHB fabrics was found after eight weeks of aging. Microscopy observation showed the time evolution process on the fine internal structure of such PLA/PHB fabrics during the experimental periods. As seen in Figure 7, the enlarged optical microscope image of the virgin PLA/PHB textile displayed an open-knitted structure with loose loops. It can be clearly seen that the weave structure of the PLA/PHB fabrics was greige-type, which was made by the single jersey-knitting manufacturing processes. Such a knitting structure benefits from being lightweight with a high surface area and is commonly applied for producing sportswear and T-shirts in the textile industry. The PLA/PHB fabrics were made by knitting multi-filament yarns. The yarns were made from multiple long fibers, which were the smallest and fundamental structural unit of the PLA/PHB fabrics. After four weeks of aging, some pitting and erosion were first evidenced at the stress-concentrated region, yet the whole integrity of the fabrics/yarns was not lost. After a longer aging time (eight weeks), the whole integrity of the PLA/PHB yarns was broken completely, and the internal interlacing connection was destroyed to a large degree. The release of (micro)fibers might have occurred from then on. The microscope image of the PLA/PHB multi-filament yarns illustrates the PLA/PHB yarns consisting of multiple long fibers, which were held together without any adhesive in a twistless process. The initial diameter of the fresh yarns was measured in the range of 190 µm to 210 µm. The diameter of fibers was averaged to be 20 µm. After seawater accessed and attracted these PLA/PHB fibers in the filament form, some fibers split from the initial yarn, whose structure became loose. Some coarse cross-sections were found at the ends of the broken PLA/PHB fibers and the diameter of these fibers was also reduced to 18 µm, as measured in Figure 8.

For comparison, the bulky PLA/PHB plastics, rather than the PLA/PHB fabrics/fibers/yarns, were also tested for their hydrolysis degradability. The chemical composition of these bulky PLA/PHB plastics was the same as that of PLA/PHB fabrics/fibers/yarns. The optical photographs of the PLA/PHB fabrics before and after immersion in seawater under different conditions for four weeks are shown in Figure 9. No visible hole or crack on the surface was identified after immersion for four weeks, and their structural integrity was well-maintained during the aging experiments. Only the surface color of the immersed samples appeared a little whiter than that of the sample without immersion. This resulted from the UV light’s bleaching effect. This result is consistent with Yu et al.’s report on the degradation behavior of transparent PLA films [28]. However, the actual mass of the bulky PLA/PHB plastics remained the same without any changes after immersion for four weeks. The results are shown in Table 1. Such phenomena were in good agreement with most previous reports questioning the practical degradability of PLA or PHB sample [16,17]. In this regard, the statement on the degradability of PLA/PHB without referring to the physical shape and dimensions was not tenable.

To further determine the effects of environmental aging factors on the mechanical behavior of bulky PLA/PHB plastics, the PLA/PHB plastics were cut into standard dogbone-like samples. As shown in Figure 10, the typical tactile curves of the PLA/PHB plastics after aging and immersion under continuous tensile loads show that the applied force initially increases linearly up to a certain extension, after which further extension requires a smaller increment in the applied force. Such behaviors were different from that of fresh and blank PLA/PHB plastics. In the latter case, there was an extra-prolonged region after the plastics region. The experimental results have also been derived from the experimental curves and summarized in Table 2. The Young’s modulus was calculated according to the slope of the initial linear portion of experimental curves and the Young’s modulus of the fresh PLA/PHB plastics was 123.74 MPa. After four weeks of immersion, all the PLA/PHB plastics showed similar rigidity, but the Young’s modulus was reduced by ~20% of the fresh PLA/PHB plastics. These behaviors revealed that the abiotic hydrolysis had a significant deteriorative effect on the mechanical properties, including both the elastic property and the plastic property, in the degradation process. Both the tensile strain and tensile strength of PLA/PHB plastics were also reduced significantly after immersion and aging. Generally, the crystalline phase plays a vital role in elasticity, whereas the non-crystalline, highly deformable amorphous phase of PLA/PHB blends affects their ductility. Thus, the hydrolysis degradation deteriorated the mechanical properties regardless of the crystalline phase or the amorphous phase of the PLA/PHB blends. Even the higher crystalline region, coupled with the lower mobility of long polymer chains, could not hinder water from diffusing into the gaps between the polymer chains. Thus, the Young’s modulus values were weakened after aging. This result was extremely different from that of PLA/PHB yarns, as described in our previous report [18]. The difference in mechanical properties resulted from the difference in the physical dimensions and/or the thermal manufacturing process of the bulky plastics and the melt-spun fibers/yarns.

The ESI-Q-TOF MS spectroscopy was applied to evaluate the distribution of various oligomeric fractions of PLA/PHB fabrics after the degradation process. These fragmentation patterns were identified through the analysis of the accurate mass-to-charge (*m*/*z*) value of fragment ions obtained from ESI-Q-TOF MS spectroscopy. As shown in Figure 11, the ESI-MS of PLA/PHB fabrics shows a series of regular signals with a peak-to-peak *m*/*z* increment of 72.00 ± 1.00 between two adjacent signals in the high *m*/*z* region of this spectrum, ranging from 2717.80 to 4591.40, which is attributed to homo-oligomer fractions generated by the successive cleavage of the ester bonds of PLA molecules. The theoretical mass of the repeating lactyl unit was calculated to be 72.02 atomic mass units. In the low *m*/*z* regions, the homo-oligomeric fractions with a characteristic repeating Δ*m*/*z* unit of 86.00 ± 1.00 between two adjacent signals could be assigned to that of PHB polymers after similar cleavage of the ester bond occurring in the following manner, as shown in Figure 11. The most intense peak in terms of relative abundance in the ESI-MS spectrum refers to blank PLA oligomers before immersion in seawater, located at *m*/*z* 3510.08. The other intense peak of PLA after four weeks of immersion and aging in static seawater, aerated seawater, UV-lighting static seawater, and UV-lighting aerated seawater was located at *m*/*z* 3294.02, *m*/*z* 3223.00, *m*/*z* 2933.91, and *m*/*z* 2861.89, respectively. Meanwhile, the maximum *m*/*z* value of the characteristic product ion peak on the ESI-MS spectrum of PLA before and after immersion in static seawater, aerated seawater, UV-lighting static seawater, and UV-lighting aerated seawater was located at *m*/*z* 4591.40, *m*/*z* 4447.35, *m*/*z* 4087.24, *m*/*z* 4015.23, and *m*/*z* 3943.21, respectively. Thus, the results revealed that the average degree of the polymerization of PLA decreases approximately from 63~64 to 54~55 after immersion and aging by assuming the polymer was detected with a single-charged molecular fraction. This illustrates that the degradability of both PLA and PHB is mainly induced by abiotic hydrolysis in South China Seawater. Therefore, the hypothesized degradation generated a wide range of homo-oligomeric fractions of PLA/PHB with smaller molecular weights, as confirmed by MSI-QTOF MS. This reduction in the molecular weights of PLA and PHB polymers gives a reasonable explanation for the adverse effects of the hydrolysis process on the mechanical properties of PLA/PHB samples in either bulky-plastics form or in the form of fibers/yarns, as mentioned previously. More importantly, another deteriorating effect derived from the variation in the crystalline degree of PLA polymers within the PLA/PHB fibers/yarns was due to their hydrophilicity. However, the change in the crystalline degree of PHB was not significant under parallel conditions because of its hydrophobicity.

## 4. Conclusions

In summary, the hydrolysis degradability of PLA/PHB products in the form of fabrics after immersion in South China Seawater was confirmed based on the mass loss of dried PLA/PHB fabrics. The degradation rate in terms of dry mass loss was applied to monitor the abiotic hydrolysis process of PLA/PHB fabrics during immersion in seawater. The effects of multiple environmental parameters involved UV irradiation and/or dissolved oxygen present in the marine environment accelerating the degradation of the PLA/PHB fabrics. Due to that the degradation rate under the four parallel conditions differed, and follows the order: (UV + Air) > (UV) >> (Air) > (static seawater). Conversely, the mass loss of bulky PLA/PHB plastics within four weeks was not detected at all, yet the variation in their mechanical properties was significant. Thus, the hydrolysis degradation of PLA/PHB was highly dependent on their physical dimensions. Such hydrolysis degradation of PLA/PHB in South China Seawater was successfully simulated and predicted by an artificial intelligence (AI) algorithm model based on a three-layered, eight-neuron topology. The results confirmed that artificial neural network modeling could not only effectively fit the experimental data but also predict the behavior of the hydrolysis degradation process under various conditions within the experimental period. Thereafter, such an AI algorithm model provides a feasible solution for the data-fitting and prediction of the degradation behavior of polymers under various marine seawater aqua environments, not limited to PLA/PHB bio-polymers.

## Figures and Tables

**Figure 1 polymers-15-00082-f001:**
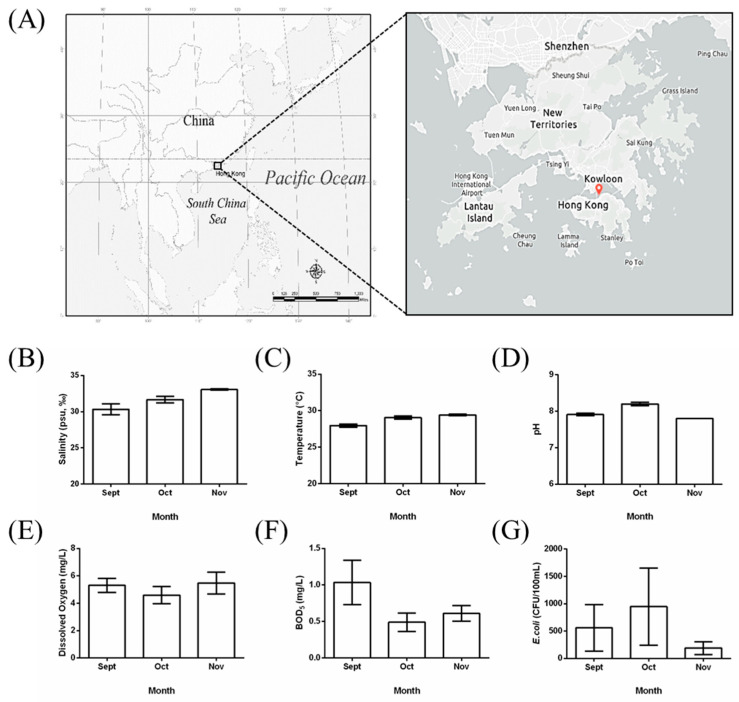
(**A**) The location of the subtropical seawater sampling site on the coastline of southern China, shown on a geographical map. The basic physicochemical properties are shown, including (**B**) the salinity, (**C**) the temperature, (**D**) the acid/base values, (**E**) the dissolved oxygen concentration, (**F**) the five-day biochemical oxygen demand (BOD_5_) values, and (**G**) the number of *E*. *coli* bacteria during the experimental period.

**Figure 2 polymers-15-00082-f002:**
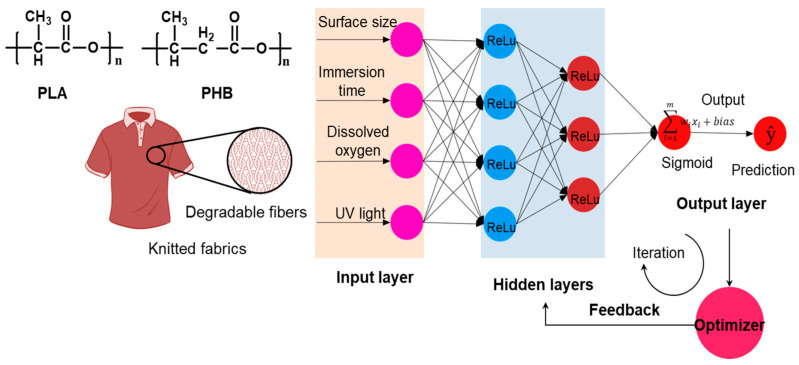
Schematic illustration showing the lifetime prediction by experimental data processing based on the artificial neural network (ANN) model.

**Figure 3 polymers-15-00082-f003:**
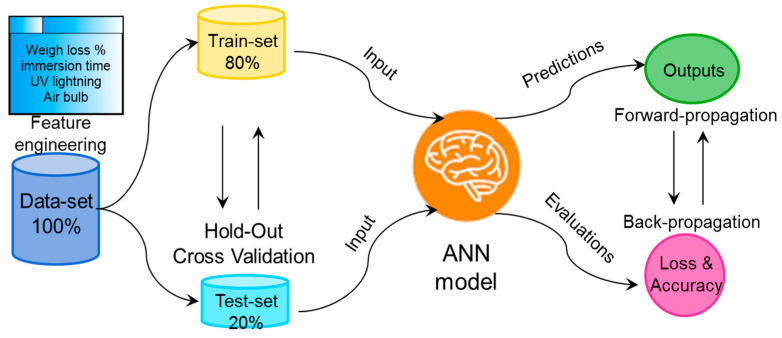
Flow chart of the proposed ANN model for predicting the degradation process of PLA/PHB fabrics.

**Figure 4 polymers-15-00082-f004:**
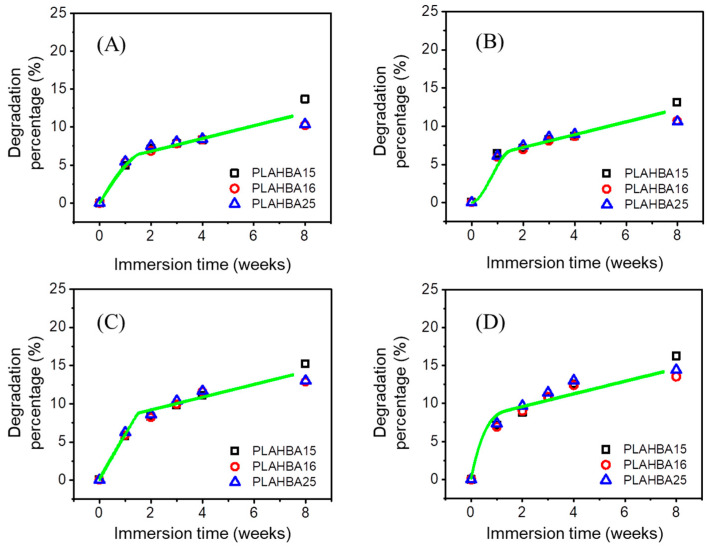
Evolution of weight loss percentage of degradable PLA/PHB fabrics under multiple environmental parameters: (**A**) static seawater, (**B**) aerated seawater, (**C**) UV-lighting static seawater, and (**D**) UV-lighting aerated seawater. Note: all the discrete scatter points represent the experimental mass-loss percentage data of PLA/PHB fabrics under static seawater and/or UV-lighting aerated seawater. The green line represents the predicted data of PLA/PHB fabrics under static seawater and/or UV-lighting aerated seawater generated by the ANN model.

**Figure 5 polymers-15-00082-f005:**
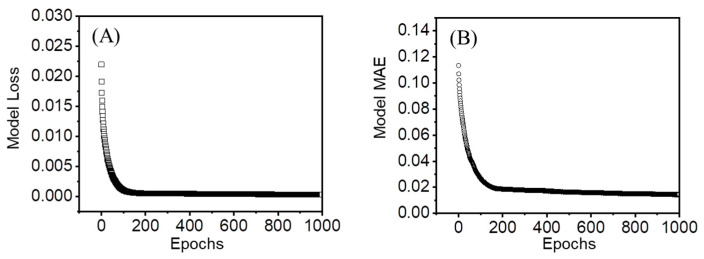
(**A**) The MSE loss function and (**B**) MAE function of the mass-loss percentage of degradable PLA/PHB fabrics under multiple environmental parameters.

**Figure 6 polymers-15-00082-f006:**
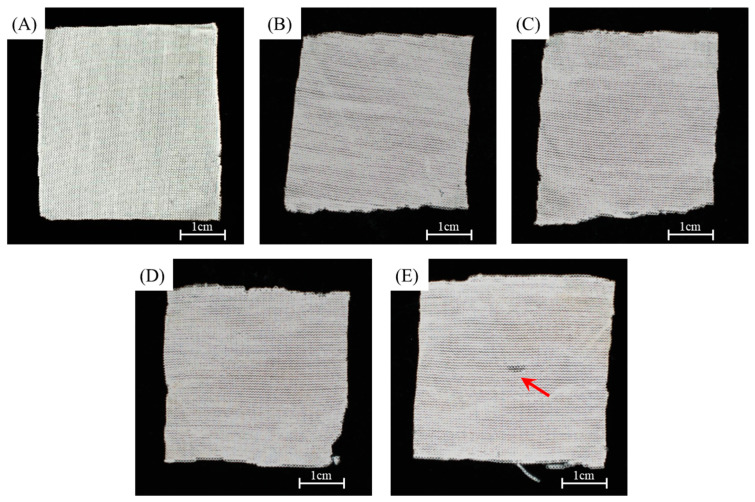
Digital images of (**A**) fresh multi-filament PLA/PHB fabrics, and multi-filament PLA/PHB fabrics after eight weeks of immersion in (**B**) static seawater, (**C**) aerated seawater, (**D**) UV-lighting static seawater, and (**E**) UV-lighting aerated seawater. Note: the red arrow indicated the broken point in the fabrics.

**Figure 7 polymers-15-00082-f007:**
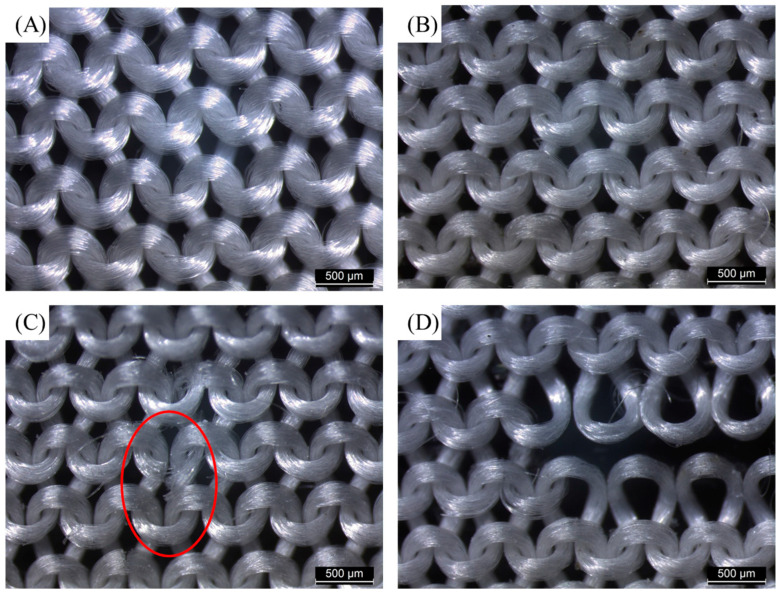
Digital microscopic images of (**A**) the initial PLA/PHB knitted fabrics and the PLA/PHB knitted fabrics after (**B**) two weeks, (**C**) four weeks, and (**D**) eight weeks exposed under both UV light and aerated seawater. Note: the red circle indicated the broken part in the yarns of the fabrics.

**Figure 8 polymers-15-00082-f008:**
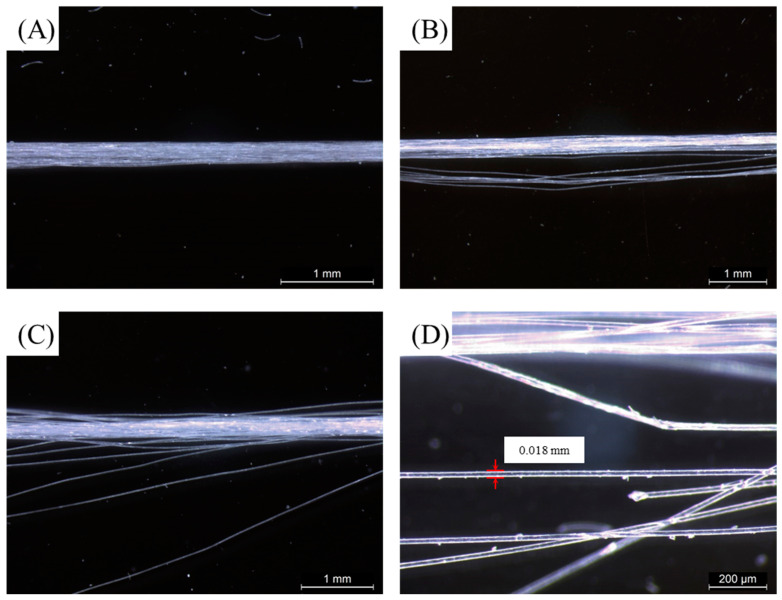
Typical microscopic images of (**A**) the initial PLA/PHB yarns and the PLA/PHB yarns after (**B**) two weeks, (**C**) four weeks, and (**D**) eight weeks immersed in both UV lighting and aerated seawater.

**Figure 9 polymers-15-00082-f009:**
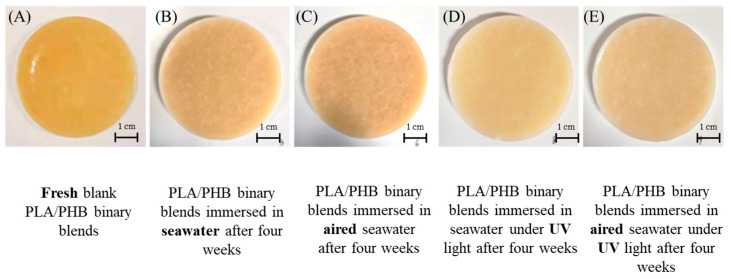
Evolution of the appearance of bulky PLA/PHB plastics before (**A**) and after immersion in seawater under multiple environmental parameters, including (**B**) static seawater, (**C**) aerated seawater, (**D**) UV-lighting static seawater, and (**E**) UV-lighting aerated seawater, for four weeks.

**Figure 10 polymers-15-00082-f010:**
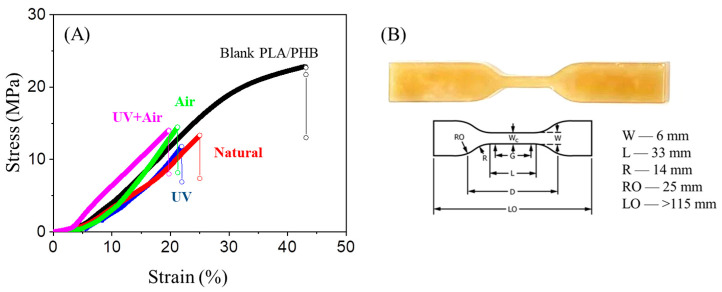
(**A**) Stress–strain curves of dogbone-like PLA/PHB plastics before and after immersion for four weeks in static seawater (Natural), aerated seawater (Air), UV-lighting static seawater (UV), and UV-lighting aerated seawater (UV + Air). (**B**) The dogbone-shaped specimens and their standard physical dimensions (ASTM D638 Standard Test Methods Type IV) for each PLA/PHB plastic were tested.

**Figure 11 polymers-15-00082-f011:**
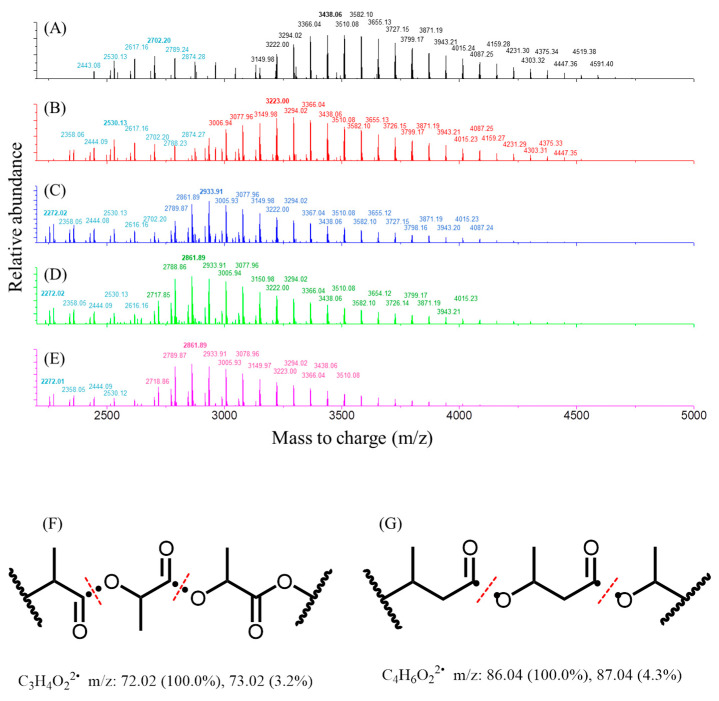
Electrospray ionization mass spectra (ESI-MS) of (**A**) fresh, blank PLA/PHB fabrics and PLA/PHB fabrics under multiple environmental parameters, including (**B**) static seawater, (**C**) aerated seawater, (**D**) UV-lighting static seawater, and (**E**) UV-lighting aerated seawater. (**F**) The MS fragmentation analysis of PLA polymer and (**G**) the MS fragmentation analysis of PHB polymer.

**Table 1 polymers-15-00082-t001:** Results of the mass weights of PLA/PHB plastics before and after four weeks’ immersion under various environmental conditions.

PLA/PHB Samples	Fresh Net Weight (g)	Net Weight after 4 Weeks (g)
Natural seawater	14.9461 ± 0.0038	14.9449 ± 0.0056
Air	14.9985 ± 0.0542	14.9942 ± 0.0552
UV light	13.6431 ± 1.8326	13.6426 ± 1.8376
Air + UV light	14.9460 ± 0.0023	14.9569 ± 0.0131

Note: no distinct weight loss was found in the bulky PLA/PHB samples.

**Table 2 polymers-15-00082-t002:** Results of the mechanical properties of the PLA/PHB samples obtained from tensile tests.

Samples	Young’s Modulus (MPa)	Tensile Strain @ Maximum Load (%)	Maximum Tensile Stress (MPa)	True Strain @ Break (Standard) (mm/mm)	True Stress @ Break (Standard) (MPa)
Blank	123.74	33.86	21.66	0.29	27.72
Natural	85.74	24.99	13.36	0.22	16.69
Air	88.32	23.03	16.46	0.21	20.25
UV	105.26	21.91	11.83	0.20	14.42
Air + UV	86.58	19.71	14.08	0.18	16.72

## Data Availability

Not applicable.

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
