# Peer review of "Evaluating and Modeling the Degradation of PLA/PHB Fabrics in Marine Water"

_polymers, 2022, doi:10.3390/polym15010082_

Round 1

Reviewer 1 Report

The authors prepared the PLA/PHB fabrics, and studied the degradation behavior of this PLA/PHB blends. The authors did a very careful job to investigate the degradation behavior of the PLA/PHB blends. However, I have several concerns about this paper:

1)     The ratio of PLA/PHB blends was 70/30, why the authors chose this ratio? What about the other ratios, such as 50/50, 30/70 et al.

2)     The SEM images of the PLA/PHB fabrics before and after degradation should be provided.

3)     Fig.2, the chemical structures of PLA and PHB should be revised.

4)     The authors mentioned “This requires ingenious balance between these two aspects” in introduction section (Page 3, Line 102), whether the degradation rate of as-prepared PLA/PHB blends could be controlled?

Author Response

Response to Editor and Reviewers’ Comments

We highly appreciate the editor and reviewers for your valuable time and effort in reviewing and commenting our manuscript. We think these suggestions are very helpful for our present and future work. We have carefully considered the all comments and revised our manuscript accordingly. All the revisions are highlighted in the revised manuscript for easy reference. The detailed comments and responses are listed as below:

Reviewer #1:

Comments: The authors prepared the PLA/PHB fabrics, and studied the degradation behavior of this PLA/PHB blends. The authors did a very careful job to investigate the degradation behavior of the PLA/PHB blends. However, I have several concerns about this paper:

1) The ratio of PLA/PHB blends was 70/30, why the authors chose this ratio? What about the other ratios, such as 50/50, 30/70 et al.

Response: Thanks for this reviewer’s comments. We have considered all reviewers’ comments and revised our manuscript carefully. It is known that both PLA and PHB including its derivatives are semicrystalline polyester polymers. In terms of the PLA/PHB binary blends, their optimized mass ratio was in the range of 20%~40% according to most literature reports. This is because that the PLA/PHB blends exhibited a remarkable improvement of phase morphology, rheological properties, tensile strength, impact resistance, modulus and max elongation at break as well as their biodegradability with their mass ratio following into this range. It might be due to the partial miscibility of PHB and PLA blends, as verified by the DSC analysis. Since their ratio value has already been recommended, it was fixed on the 70/30 in this work. This is also because that other ratio like 50/50 might not be applicable for the melt spinning into filaments/fibers. According to the reviewer’s concern, the Introduction section, the Experimental section and reference citation of the manuscript has been revised (See Page 2 of 16, Page 3 of 16).

2) The SEM images of the PLA/PHB fabrics before and after degradation should be provided.

Response: Thanks for the professional advice. Indeed, the SEM analysis could provide higher resolution and local microscopic morphology images. However, the PLA/PHB fabrics were electrical insulator, which need to be coated with a noble metal layer or carbon layer. Such a conductive layer will affect the microscopic morphology analysis results. Furthermore, under the bombardment of high energy electron beams, the PLA/PHB samples might melt due to the thermal effect, which might pollute the SEM instrument. It seems that environmental SEM analysis might be much more suitable in this case. However, the observation depth of the 3D structure of the PLA/PHB fabrics was so high that the electron beams could not be focused on a plat plane. This resulted in the difficulty in the focus action of electron beams of SEM, which could not make a clear SEM image. Taking account of all these considerations, in this case, we have chosen the optical microscopy as the analysis tool, which provided the sufficient clear and global results of the PLA/PHB fabrics on the micrometer scale as shown in Fig.7 and Fig.8.

3) Fig.2, the chemical structures of PLA and PHB should be revised.

Response: Thanks a lot for the reviewer’s correction. According to the reviewer’s suggestion, we redraw Fig.2 in the revised manuscript (See Page 5 of 16).

4) The authors mentioned “This requires ingenious balance between these two aspects” in introduction section (Page 3, Line 102), whether the degradation rate of as-prepared PLA/PHB blends could be controlled?

Response: Thanks for the reviewer’s valuable concern. To our best knowledge, the degradation rate of PLA/PHB blends could not be controlled currently. However, it’s a splendid suggestion/idea to design the novel and bio-safe PLA/PHB fabrics with the controllable degradation if possible. Some high impact paper has been published on top journals like J. Am. Chem. Soc. (https://pubs.acs.org/doi/10.1021/jacs.1c07508). In order to encourage/inspire the following research in this respect, we cited this paper in the revised manuscript (See the last paragraph of Introduction section, Page 3 of 16), as the reviewer mentioned and proposed.

Author Response

Response to Editor and Reviewers’ Comments

We highly appreciate the editor and reviewers for your valuable time and effort in reviewing and commenting our manuscript. We think these suggestions are very helpful for our present and future work. We have carefully considered the all comments and revised our manuscript accordingly. All the revisions are highlighted in the revised manuscript for easy reference. The detailed comments and responses are listed as below:

Reviewer #2:

Comments:

Manuscript number: polymers-2092517

Title: Evaluating and Modeling the Degradation of PLA/PHB Fabrics in Marine Water

Authors: Qi Bao, Ziheng Zhang, Heng Luo, Xiaoming Tao

Recommendation: Minor revision

Major comments:

  1. Page 3, lines 133-134. Are the oxygen and UV levels used in the experiment comparable with that in the ocean? If not, the result may not be practical. It is suggested that the authors give some discussion on this.

Response: Thanks for the reviewer’s professional concern. Actually, the complexity of UV level was by far complicated than what we thought. This is because the wavelength, the output power and time duration of UV explosion will affect the degradation process. Currently, we could use the light of the Xenon lamp to simulate the natural sunlight. In this case, we use commercial Hg lamp with fixed exposure time and output power to investigate the aging effects of UV light in order to simplify the mathematical data feature process. This process will involve the calculation of both exposure time evolution and wavelength evolution. Furthermore, this will also involve the absorption and reflection of PLA/PHB against UV with different wavelength. More importantly, as the reviewer’s concern, the practical situation was much more complicated in reality. This is because it involved at least two phases including the seawater and marine atmosphere. The aging effects of UV will also vary in different depth of seawater as a result of the absorption and scattering of seawater. Obviously, this progress will also result in the trouble in the measurement and results validation of UV level. I am not sure whether the top supercomputer in the world could give some mathematics result in this respect. In this work, we use logic numbers 1 and 0 to represent the cases with and without UV for simplicity; given the complexity of UV level was out of the range of this work. Regarding the oxygen level, the average saturated dissolved oxygen value of the surface seawater in sampling site in 2020 was in the range of 7.0-7.1 mg/L. The average saturation percentage of dissolved oxygen in 2020 was 79%. In this work, the dissolved oxygen level was almost saturated by supplying air with the flow rate of 4.3 ± 1.0 SLPM. According to the reviewer’s suggestion, we modified the discussion section in the revised manuscript ( See Page 5 of 16).

  1. Page 7, Figure 4. Could the authors explain what are PLAHBA 15/16/25? Could you also show an overlay or data table? The difference between Figure 4A vs 4B, 4C vs 4D are not so obvious from the current figures.

Response: Thanks a lot for the reviewer’s concern. Sorry for such a misunderstanding. This is because we use three parallel PLA/PHB fabric samples with different physical size (3 cm×5 cm, 4 cm×4 cm and 5 cm×5 cm). Then, the according PLA/PHB fabrics samples were labeled as PLAHBA15, PLAHBA16, and PLAHBA25, respectively. We have modified the description in revised manuscript (See Page 5 of 16). For clarity, we attached all the raw experimental and predicted data of PLA/PHB fabrics in SI-1. Thanks to the reviewer’s suggestion, we agreed with them. The difference between A) static seawater and B) aerated seawater was not obvious. Similarly, the difference between C) UV lighting static seawater and D) UV lighting aerated seawater was not obvious as well. Thereafter, we modified the according discussion and description in the revised manuscript (See Page 7 of 16).

  1. Page 10, lines 346-347. The authors mentioned the behavior is quite different from the PLA/PHB yarns, it is suggested to add the data of the yarns in SI.

Response: Thanks a lot for the reviewer’s concern. In this work, we focused on the comparison between the bulky PLA/PHB plastics and the PLA/PHB fabrics. Thereafter, this sentence has been adjusted in the revised manuscript for clarity (See Page 10-11 of 16). The mechanical behaviors of PLA/PHB yarns have already been described in the previous publication (Polymers 2022, 14, 1216 https://doi.org/10.3390/polym14061216). This reference has been cited in the revised manuscript.

  1. Page 12, lines 395-397. ESI-MS suffers a lot from competitive ionization effect, where the ionization of high MW species is suppressed by low MW species. For a more accurate average MW and PDI evaluation, HPLC-CAD, HPLC-ELSD, GPC are likely better options, but this is all up to the authors decision and instrument availability.

Response: Thanks a lot for the reviewer’s professional advice by providing some advanced techniques. Currently, the HPLC-CAD and HPLC-ELSD have not found in the local region. In our department, the Waters GPC instrument still works. However, the columns need to be adjusted and activated, which needs 2-4 months to order a new specific column from USA Instrument Company. And the disadvantage of GPC was that we could not identify the multiple peaks to be that of PLA or PHB or other impurity. The ESI-MS was easy handled by comparison in particular the TOF-MS one. Thereafter, thanks to the reviewer’s kind suggestion, these techniques might be adopted in our future work.

  1. Page 10, lines 346-347. The authors mentioned the behavior is quite different from the PLA/PHB yarns, it is suggested to add the data of the yarns in SI.

Response: Thanks a lot for the reviewer’s concern. In this work, we focused on the comparison between the bulky PLA/PHB plastics and the PLA/PHB fabrics. Thereafter, this sentence has been adjusted in the revised manuscript for clarity (See Page 10-11 of 16). The mechanical behaviors of PLA/PHB yarns have already been described in the previous publication (Polymers 2022, 14, 1216 https://doi.org/10.3390/polym14061216). This reference has been cited in the revised manuscript.

Minor comments and corrections:

  1. Page 2, line 80. “except from” should be “except for”.

Response: Thanks for pointing this mistake for us. This mistake has been corrected in the revised manuscript (See 2nd paragraph, page 2 of 15).

  1. Page 1, line 33. Page 2, line 85, etc. “accumulation for decades. [3]” should be “accumulation for decades [3].”

Response: Thanks for pointing these format mistakes for us. All the format mistakes throughout the manuscript have been corrected, accordingly.

  1. Page 3, lines 140-144. The authors should add the instrument vendor name, and country.

Response: Thanks for pointing this for us. According to the editor’s requirement, the vendor name and country of instruments and software has been added in the revised manuscript (See 2. Materials and Methods section, page 3 and 4 of 16).

  1. Page 4, line 151. The full name of ANN should be used when mentioning it for the 1st time.

Response: Thanks for pointing this for us. For clarity, the full name (artificial neural network) of ANN has been rewritten in both Page 3 and Page 4 of 16.

  1. Page 5, Figure 1. Seems like the 1st sentence of figure caption should be removed. And do you have data showing the UV level at the sampling site?

Response: Thanks for pointing this for us. The first sentence of Figure caption of Fig. 1 has been removed in the revised sentence (See page 5 of 16). Regarding the UV level, the complexity of UV level was by far complicated than what we thought as mentioned above. In this work, we described the type of UV light and rate output power of UV light in the Experimental section (See Section 2.1 Page 3 of Page 16). Thereafter, the traditional UV index data has not been added in this work in order to avoid the misunderstanding.

  1. Page 6, line 214. “salient their capability …” “their” should be removed.

Response: Thanks for pointing this mistake for us. We corrected this sentence in the revised manuscript (See the first paragraph in page 6 of 16).

  1. Page 8, line 272. “founded” should be “found”.

Response: Thanks for pointing this grammar mistake for us. Sorry about that. We corrected this word into the right one in the revised manuscript (See the first paragraph in page 8 of 16).

  1. Page 9, line 308. “in Fig. 7” should be “in Fig. 8”; Line 318, “Fig. 8” should be “Fig. 9”.

Response: Thanks for pointing these two mistakes for us. Both the Fig. 7 and Fig.8 have been corrected in the revised manuscript (See Page 9 of 16).

  1. Page 10, line 334. “Figure 9” should be “Figure 10”.

Response: Thanks for pointing this mistake for us. The Fig. 9 has been corrected to Fig. 10 in the revised manuscript (See Page 11 of 16).

  1. Page 11, Figure 9. Figure caption needs to be corrected.

Response: Thanks for pointing this for us. The Figure caption of Fig. 9 has been corrected in the revised manuscript (See Page 11 of 16).

  1. Page 12, line 390. The extra “m/z 2861.89” should be removed.

Response: Thanks for pointing this redundancy error for us. The extra “m/z 2861.89” has been deleted in the revised manuscript (See Page 12 of 16).

  1. Page 13, Figure 11. Missing figure captions.

Response: Thanks for pointing this for us. The Figure caption of Figure 11 has modified accordingly in the revised manuscript (See Page 13 of 16).

Round 2

Reviewer 1 Report

I suggested that this paper can be accepted in present form.